# The Cardioprotective Potential of Herbal Formulas in Myocardial Infarction-Induced Heart Failure through Inhibition of JAK/STAT3 Signaling and Improvement of Cardiac Function

**DOI:** 10.3390/ph17091132

**Published:** 2024-08-27

**Authors:** Youn-Jae Jang, Hye-Yoom Kim, Se-Won Na, Mi-Hyeon Hong, Jung-Joo Yoon, Ho-Sub Lee, Dae-Gill Kang

**Affiliations:** 1Hanbang Cardio-Renal Syndrome Research Center, Wonkwang University, Iksan 54538, Republic of Korea; j8626@naver.com (Y.-J.J.); hyeyoomc@naver.com (H.-Y.K.); sewon3066@naver.com (S.-W.N.); mihyeon123@naver.com (M.-H.H.); mora16@naver.com (J.-J.Y.); 2College of Oriental Medicine, Professional Graduate School of Oriental Medicine, Wonkwang University, Iksan 54538, Republic of Korea

**Keywords:** korean herbal formulas, dohongsamul-tang, myocardial infarction, heart failure, cardiac function, cardioprotection

## Abstract

Myocardial infarction (MI) is a leading cause of heart failure, characterized by adverse cardiac remodeling. This study evaluated the cardioprotective potential of Dohongsamul-tang (DHT), a traditional Korean herbal formula, in a rat model of MI-induced heart failure. Rats underwent left anterior descending (LAD) artery ligation and were treated with either 100 mg/kg or 200 mg/kg of DHT daily for 8 weeks. DHT treatment significantly improved cardiac function, as evidenced by increased ejection fraction (EF) from 62.1% to 70.1% (100 mg/kg) and fractional shortening (FS) from 32.3% to 39.4% (200 mg/kg) compared to the MI control group. Additionally, DHT reduced infarct size by approximately 63.3% (from 60.0% to 22.0%) and heart weight by approximately 16.7% (from 3.6 mg/g to 3.0 mg/g), and significantly decreased levels of heart failure biomarkers: LDH was reduced by 37.6% (from 1409.1 U/L to 879.1 U/L) and CK-MB by 47.6% (from 367.3 U/L to 192.5 U/L). Histological analysis revealed a reduction in left ventricle (LV) fibrosis by approximately 50% (from 24.0% to 12.0%). At the molecular level, DHT inhibited the expression of phospho-JAK by 75% (from 2-fold to 0.5-fold), phospho-STAT3 by 30.8% (from 1.3-fold to 0.9-fold), Bax/Bcl-2 by 56.3% (from 3.2-fold to 1.4-fold), and caspase-3 by 46.3% (from 1.23-fold to 0.66-fold). These results suggest that DHT exerts cardioprotective effects by modulating the JAK/STAT3 signaling pathway, highlighting its potential as a therapeutic option for heart failure.

## 1. Introduction

Dohongsamul-tang (DHT), a renowned traditional herbal formula used in Korean medicine, is composed of six medicinal herbs: *Angelica gigas* Nakai, *Prunus persica* Batsch, *Rehmannia glutinosa* (Gaertn.) DC., *Paeonia lactiflora* Pallas, *Cnidii officinale* Makino, and *Carthamus tinctorius* Linné (Table 1). Historically, DHT has been employed in the treatment of irregular menstrual disorders, migraines, and immune disorders, reflecting its broad therapeutic applications in traditional medicine [1,2,3].

Recent studies have identified several bioactive compounds in DHT that contribute to its therapeutic effects [4]. These include (+)-catechin, epicatechin, 4-hydroxy cinnamic acid, and rutin, which have demonstrated antioxidant and anti-inflammatory properties [4,5,6,7]. Additionally, glycyrrhizin and liquiritin apioside, present in DHT, are known for their roles in modulating inflammatory responses and protecting against oxidative stress [8,9]. Myricetin and protocatechuic acid also contribute to the cardioprotective effects through their ability to reduce oxidative damage and improve endothelial function [10,11].

These compounds are believed to work synergistically to enhance cardiovascular health and mitigate cardiac damage. However, despite its widespread usage, the precise mechanisms underlying the cardioprotective effects of DHT on cardiac function remain elusive. To address this gap in knowledge, our study was designed to investigate the impact of DHT on myocardial infarction (MI)-induced heart failure. Our current study aims to explore whether DHT has a specific potential for treating heart failure caused by MI. While recent studies, such as those by Wang et al. [12] and Xie et al. [13], have highlighted the cardiovascular benefits of similar herbal formulations, the precise mechanisms by which DHT exerts its cardioprotective effects, particularly in the context of MI, remain unclear. By focusing on this critical area, our research seeks to provide valuable insights into how DHT can be utilized as a therapeutic intervention for MI-induced cardiac dysfunction.

MI, a leading cause of ischemic heart disease, occurs when one of the coronary arteries becomes blocked, resulting in a heart attack. This event induces severe disruptions in cardiac function, ultimately leading to reduced heart function and potentially fatal outcomes [14,15]. MI is also associated with the development of abnormal heart rhythms, and it frequently leads to adverse cardiac remodeling and dysfunction, culminating in heart failure [16]. Animal models of MI are commonly employed to study the progressive myocardial changes that occur over an extended period of time [17]. By inducing coronary ligation in rats, we can closely replicate the pathophysiological alterations observed in human ischemic heart disease [18]. Among the various surgical methods used to induce ischemia, ligation of the left anterior descending artery (LAD) is the most commonly performed technique [19,20]. In the MI animal model, the extent of the infarcted area serves as a representative measure of heart failure and may act as a prognostic factor for complications following left ventricular (LV) remodeling. Moreover, heart failure resulting from MI is characterized by left ventricular end-systolic (LVSD) systolic dysfunction due to aberrant signaling within the systolic mechanism [21]. This compromised contractility leads to a decrease in stroke volume (SV) due to incomplete ventricular output, resulting in increased left ventricular end-diastolic volume and pressure (LVEDP) [22]. Consequently, a substantial reduction in ejection fraction (EF) and fractional shortening (FS) occurs alongside the decline in both EF and FS and the maximum slope of the end-systolic maximal peak rate of pressure development, indicative of significant cardiac functional deterioration in response to MI [23,24]. Ventricular remodeling, directly implicated in post-infarction ventricular dilation, can profoundly influence LV function and impact long-term survival outcomes [25]. Additionally, the accumulation of collagen in the heart, leading to fibrosis, is widely accepted as a characteristic feature of heart disease, contributing to both diastolic and systolic dysfunction as well as increased susceptibility to cardiac arrhythmias [26,27].

Given the suitability of the MI animal model for studying the effect of DHT on heart failure, we generated this model to confirm the potential of DHT in ameliorating heart failure. Our study aims to determine whether DHT exhibits a protective effect against cardiac dysfunction and fibrosis in rats with heart failure resulting from myocardial infarction. In numerous studies, the janus kinase (JAK) signal transducer and activator of transcription 3 (STAT3) signaling pathway have been implicated in the pathogenesis of cardiovascular remodeling following MI [28]. This signaling cascade is capable of transducing intracellular signals from various cytokines, thereby regulating several physiological and pathophysiological processes [29]. Under stressful conditions such as myocardial ischemia, the JAK/STAT3 pathway is known to play a crucial role and is associated with the upregulation of caspase-3 activity and Bcl-2 antagonist X (Bax) expression, ultimately leading to apoptosis [30]. However, the potential modulation of MI-induced myocardial dysfunction by DHT through the JAK/STAT3 signaling pathway has not been investigated. Therefore, our study aims to explore the potential relationship between the cardioprotective actions of DHT and JAK/STAT3 signaling.

## 2. Results

### 2.1. Chemical Characterization and Composition of DHT

To determine the bioactive components of DHT, a comprehensive chemical characterization was performed using various analytical techniques, including retention time analysis, exact mass spectrometry (MS), and MS/MS fragmentation analysis. A total of six compounds were tentatively identified. Among these, glucose, phenylalanine, stachyose, sucrose, tryptophan, and valine were detected. While glucose and sucrose are present in DHT, they are generally not considered bioactive components in the context of therapeutic efficacy. The prominent bioactive constituents identified from the individual herbs comprising DHT include nodakenin and decursin from Angelica gigas Nakai, amygdalin from Prunus persica Batsch, catapol from Rehmannia glutinosa (Gaertn.) DC., oxypaeoniflorin and paeoniflorin from Paeonia lactiflora Pallas, ferulic acid from Cnidium officinale Makino, and safflomin A from Carthamus tinctorius Linné (Table 2). These herbs have a well-established history of clinical use for enhancing blood circulation and treating cardiovascular disorders such as hypertension and angina. Further, employing UPLC/QE analysis, we confirmed the presence of the specific bioactive components amygdalin, catapol, decursin, ferulic acid, nodakenin, oxypaeoniflorin, paeoniflorin, and safflomin A [4].

### 2.2. Effect of DHT on Infarct Size and Electrocardiogram Changes in MI-Induced Heart Failure Rats

The electrocardiogram (ECG) is a vital diagnostic tool for monitoring and detecting signals of MI. To assess the degree of heart failure manifestation caused by MI, we monitored heart function, as shown in Figure 1A and Table 3. Compared to the sham group, the MI model group exhibited increased ST segment elevation, confirming the successful induction of heart failure by MI. Triphenyl tetrazolium chloride (TTC) staining, an important index for assessing MI injury (Figure 1B), revealed an increased infarct size in the MI group compared to the sham group. However, treatment with DHT significantly reduced the infarct size (Figure 1B,C). Additionally, DHT treatment inhibited cardiac hypertrophy in MI-induced heart failure, as demonstrated in Figure 1D–F.

### 2.3. Effects of DHT on Hematology and Serum Biochemical Analysis in MI-Induced Heart Failure Rats

Hematological and serum biochemical analyses provide crucial information for assessing MI-induced heart failure. MI leads to elevated levels of myocardial-associated isoenzymes creatine kinase myocardial band (CK-MB) and lactate dehydrogenase (LDH) in the serum. Our results showed a significant increase in LDH (Figure 2A) and CK-MB (Figure 2B) levels in MI rats compared to the sham group, while these levels were significantly decreased in the DHT-treated group. Thus, DHT administration demonstrated an improvement in MI-induced heart failure.

### 2.4. Effect of DHT on Cardiac Dysfunction in MI-Induced Heart Failure Rats

Echocardiography, a versatile imaging modality, was employed to assess cardiac function parameters and manage MI-induced rat models. Echocardiographic M-mode images of MI-induced rat hearts confirmed the beneficial effects of DHT in improving the internal diameter of the chamber, which widens during MI-induced heart failure (Figure 3A). MI-induced rats exhibited significant reductions in ejection fraction (EF, Figure 3B) and fractional shortening (FS, Figure 3C) compared to the sham group. However, treatment with DHT improved cardiac dysfunction parameters, including left ventricular end-diastolic volume (LVEDV, Figure 3D), left ventricular internal diameter at end-diastole (LVIDd, Figure 3E), left ventricular end-systolic volume (LVESV, Figure 3F), and left ventricular internal diameter at end-systole (LVIDs, Figure 3G). These results indicate that DHT administration improves cardiac remodeling and dysfunction following myocardial infarction.

### 2.5. Effect of DHT on Hemodynamic Parameters in MI-Induced Heart Failure Rats

The Millar pressure-volume (PV) conductance catheter system was used to evaluate LV performance and obtain LV pressure-volume relationships (PV loop). PV loop analysis revealed typical heart failure symptoms, including increased volume and decreased pressure, in the MI group compared to the control group. However, these symptoms were ameliorated with DHT treatment (Figure 4A). The hemodynamic parameters stroke work (SW, Figure 4B), cardiac output (CO, Figure 4C), ejection fraction (EF, Figure 4D), stroke volume (SV, Figure 4E), and heart rate (HR, Figure 4F) confirmed the improved hemodynamics in MI-induced rats following DHT treatment.

### 2.6. Effect of DHT on Histopathological Changes in MI-Induced Heart Failure Rats

Histological analysis was performed to detect and quantify collagen deposition in the LV tissue of MI-induced rats. Masson’s trichrome staining (blue staining for collagen fibers, Figure 5A) and picrosirius red staining (red staining for collagen fibers, Figure 5B) were used to measure fibrosis in the LV tissue. The MI group exhibited increased collagen deposition in the left ventricle, while DHT treatment showed improvement in fibrosis. These results suggest that DHT administration ameliorates cardiac fibrosis.

### 2.7. Effect of DHT on JAK/STAT3 Signaling in MI-Induced Heart Failure Rats

To gain insights into the cardioprotective effects of DHT, we investigated the myocardial JAK/STAT3 signaling pathway at the molecular level. The MI-induced heart failure rats showed increased expression of JAK/STAT3 signaling parameters in left ventricular tissue, which significantly exacerbated myocardial apoptosis. However, treatment with DHT suppressed the expression of phospho-JAK, phospho-STAT3, Bax/Bcl-2, and caspase-3 proteins (Figure 6). These findings indicate the beneficial effects of DHT through the modulation of JAK/STAT3 signaling.

## 3. Discussion

DHT is a traditional herbal formulation documented in ancient Korean medical texts such as “Donguibogam” and “Bangyaghabpyeon”. It consists of six medicinal herbs, including *Angelica gigas* Nakai, *Prunus persica* Batsch, *Rehmannia glutinosa* (Gaertn.) DC., *Paeonia lactiflora* Pallas, *Cnidii officinale* Makino, and *Carthamus tinctorius* Linné (Table 1) [1,2,3]. DHT has been traditionally used for alleviating symptoms associated with irregular menses disorder in women and treating various conditions such as immunological disorders, migraine, hypertension, and angina [1,2,3]. However, the specific effects of DHT in the context of MI have remained unclear. Therefore, the aim of this study was to investigate whether DHT treatment could attenuate cardiac dysfunction in rats with MI-induced heart failure.

To mimic human ventricular remodeling and the subsequent development of heart failure, animal models of MI induced by surgical ligation of the LAD artery in rats were employed [31]. The MI model used in our study involves LAD artery ligation to induce a localized and acute blockage of blood flow. This approach is designed to replicate the effects of sudden and severe coronary occlusion, providing insights into acute myocardial damage and subsequent heart failure. However, the surgical procedure for inducing MI in rats can be complex and challenging, often resulting in high mortality rates [32,33]. Therefore, it was crucial to confirm the success of the surgical procedure by assessing characteristic ECG abnormalities indicative of myocardial ischemia [34]. In our study, we confirmed the occurrence of arrhythmias during surgery through ECG measurements (Figure 1A). Various parameters were evaluated after eight weeks of LAD ligation to validate the development of heart failure following MI [35].

In contrast, myocardial infarction due to atherosclerosis typically results from a gradual and progressive narrowing of the coronary arteries due to plaque buildup [36]. This type of obstruction is more chronic and diffuse, leading to partial or intermittent blood flow reduction before complete blockage occurs. Atherosclerosis-induced MI often affects multiple regions of the heart over time and may not present as a single, acute infarct [37]. While the LAD ligation model effectively simulates the immediate consequences of severe coronary obstruction, it does not fully replicate the chronic, progressive nature of atherosclerosis. To model atherosclerosis more closely, additional approaches that simulate gradual arterial stenosis would be required.

In our study, infarct size, which is frequently proportional to left ventricular remodeling and hemodynamic dysfunction, was examined [38]. Consistent with previous studies, significant increases in the infarct area, heart weight, and heart weight-to-body weight ratio were observed in the MI group compared to the sham group, confirming the successful induction of heart failure through surgery [39,40]. Echocardiography and PV loop analysis were employed to comprehensively evaluate cardiac function and remodeling [35,41,42].

Parameters such as LVEDV, LVESV, LVIDd, and LVIDs were assessed. Treatment with DHT improved these echocardiographic parameters, indicating the amelioration of cardiac remodeling and dysfunction following myocardial infarction.

Myocardial injury resulting from MI leads to the release of key myocardial enzymes, such as LDH and CK-MB, into the bloodstream. These enzymes are critical biomarkers that reflect the extent of myocardial damage, as their elevated levels are directly associated with the severity of cardiac injury [43,44]. In our study, DHT treatment significantly inhibited the levels of CK-MB and LDH, suggesting that DHT exerts a protective effect against MI-induced myocardial injury. This reduction in enzyme levels implies that DHT may help preserve the structural integrity of cardiac tissue, thereby mitigating the extent of damage following MI.

Additionally, cardiac arrhythmias are a common and dangerous complication in the context of MI-induced heart failure. These arrhythmias are strongly associated with increased myocardial fibrosis, which disrupts the normal electrical conduction pathways within the heart. Ventricular enlargement, often a consequence of heart failure, further exacerbates the risk of arrhythmias by altering the geometric and electrical properties of the heart [45]. Collagen deposition and fibrosis are key processes in the pathological remodeling of the myocardium following MI, contributing to stiffening of the cardiac tissue and impaired contractility [46,47]. Our results demonstrated that DHT treatment effectively reduced the collagen ratio in the infarcted myocardium. This reduction in collagen deposition indicates that DHT has the potential to attenuate fibrosis, thereby preserving cardiac function and reducing the risk of arrhythmias and other complications associated with heart failure [48,49].

The JAK/STAT pathway plays a pivotal role in the regulation of the mitochondrial apoptotic system, mediating both anti-apoptotic and pro-apoptotic functions depending on the context of the signaling environment [50]. In the setting of MI-induced heart failure, there is a pronounced upregulation of JAK and STAT phosphorylation, which is accompanied by a detrimental shift in the balance of apoptotic regulatory proteins. Specifically, there is a downregulation of the anti-apoptotic protein Bcl-2 and an upregulation of pro-apoptotic proteins such as Bax and caspase-3 [51,52]. This imbalance leads to increased apoptosis of cardiomyocytes, exacerbating cardiac dysfunction and contributing to the progression of heart failure. In our study, DHT treatment significantly inhibited the expression of Bax and caspase-3 while promoting the expression of Bcl-2. This modulation of the apoptotic pathway by DHT resulted in a reduced rate of cardiomyocyte apoptosis, thereby ameliorating cardiac dysfunction in rats with MI-induced heart failure. These findings suggest that DHT may exert its cardioprotective effects, at least in part, through the inhibition of the JAK/STAT pathway and the preservation of cellular survival mechanisms in the heart.

In summary, DHT, a traditional herbal formulation documented in ancient Korean medical texts, has demonstrated promising potential for improving cardiac dysfunction in a rat model of MI-induced heart failure. This study highlights that DHT has significant protective effects against myocardial injury, fibrosis, and apoptosis, with notable benefits observed in cardiac remodeling and function. These results underscore the potential of DHT for mitigating cardiac damage and improving outcomes in heart failure.

However, this study also identified some limitations. Specifically, the treatment with 200 mg/kg of DHT did not lead to the anticipated reduction in caspase-3 expression and was associated with elevated Bax/Bcl-2 ratios and p-STAT/STAT levels, as well as increased CK-MB levels. These findings suggest that high doses of DHT may have complex effects on apoptotic pathways and cardiac stress responses, indicating that the optimal dose for therapeutic efficacy needs further investigation.

Overall, despite these limitations, this study provides valuable insights into the therapeutic effects of DHT, particularly its potential to protect against myocardial injury and improve cardiac function. Future research is necessary to refine dosing strategies, determine the optimal concentration for clinical use, and fully elucidate the molecular mechanisms involved to maximize the clinical potential of DHT.

## 4. Materials and Methods

### 4.1. Preparation of DHT and UPLC/QE Orbitrap MS Analysis of DHT Compounds

DHT was prepared by boiling a mixture of 60 g each of Angelicae gigantis Radix, Persicae semen, and Rehmanniae radix Recens, along with 30 g each of Paeoniae radix, Cnidii rhizoma, and Carthami flos, in 4000 mL of distilled water (DW) for 2 h. The extract was concentrated using a vacuum evaporator, dried, and stored at −70 °C, yielding 23.2% of the original weight. The chemical constituents of DHT were analyzed using UPLC/QE Orbitrap MS in both negative and positive ion modes. For the analysis, 0.1 g of the DHT sample was extracted with 750 µL of 70% methanol. The resulting supernatant was diluted with 70% acetonitrile. Chromatographic separation was conducted using a UPLC/QE Orbitrap MS system. Key bioactive compounds, including (+)-catechin, 4-hydroxy cinnamic acid, 5-HMF, epicatechin, glycyrrhizin, liquiritin apioside, myricetin, protocatechuic acid, and rutin, were confirmed using authentic standards. We would also like to note that the DHT used in our study is consistent with that described in the recent publication by Hong et al. [4], where DHT was shown to inhibit cardiac remodeling and fibrosis through calcineurin/NFAT and TGF-β/Smad2 signaling pathways in cardiac hypertrophy [4]. This additional reference supports the validity of the DHT preparation used in our study and further corroborates its therapeutic effects.

### 4.2. Myocardial Infarction-Induced Heart Failure Animal Model

The experiment was conducted on male Sprague-Dawley rats (Koatech, Pyeongtaek, Korea) following a week of adjustment. The rats were fed a normal rodent chow diet with ad libitum access to DW and were housed under constant temperature and relative humidity conditions with a regular light/dark schedule (12 h light, 12 h dark). MI procedures were performed on rats approximately 7 weeks old with a body weight of 200–250 g. The animals were anesthetized with 4% isoflurane, intubated, and mechanically ventilated (Harvard Apparatus, Small Animal Ventilator, Boston, MA, USA) at a rate of 80–90 cycles per minute with a tidal volume of 1–2 mL/100 g. Subsequently, a diagonal incision was made in the skin over the rib cage, muscles were bluntly dissected, and a thoracotomy was performed at the 5th intercostal rib space. The MI model was generated by ligation of the LAD using a 6–0 suture, which was visualized by immediate blanching of the myocardium. Sham animals underwent identical procedures without ligation of the LAD. After the operation, the animals were monitored daily for recovery. DHT was orally administered to the animals for eight weeks, starting the day after the surgery, while the sham group received only DW. The groups were composed as follows: Sham (DW, n = 8), MI + Vehicle (DW, n = 10), MI + DHT (100 mg/kg/day dissolved in DW, n = 9), and MI + DHT (200 mg/kg/day dissolved in DW, n = 9). This study was approved by the Institutional Ethics Committee for Animal Experimentation of Wonkwang University (WKU20–23).

### 4.3. Assessment of Electrocardiography

ECG was continuously recorded using standard 3-lead skin electrodes, with two electrodes placed towards the heart on the right and left forelimbs and a neutral third electrode on the hind limb facing the heart. The ECG signals were measured using electrodes connected to a data acquisition system, Power Labs (AD Instruments, Sydney, Australia).

### 4.4. Assessment of Cardiac Injury Biomarkers

The plasma levels of LDH and CK-MB were measured using an automated clinical chemistry analyzer (FUJI DRI-CHEM NX700, FUJIFILM Corporation, Tokyo, Japan).

### 4.5. Determination of Myocardial Infarct Size

The cardiac infarct area was evaluated using 2,3,5-triphenyltetrazolium chloride dye (TTC solution 2%, Solarbio, Beijing, China) staining. The heart was kept at −20 °C for 15 min and then sliced into five 1 mm sections parallel to the coronary sulcus. All slides were incubated with PBS with 2% TTC at 37 °C for 15 min in the dark. The infarct size was expressed as a percentage of the area at risk over the total ventricular area.

### 4.6. Assessment of Cardiac Function via Echocardiography

Echocardiography was performed using an ultrasound unit with an 18LS probe set at a frequency of 14 MHz (VINNO6, Vinno Corporation, Shanghai, China). The animals were anesthetized with 4% isoflurane mixed with oxygen. They were placed in a nose cone on a thermal mat to maintain anesthesia, and during the measurement period, isoflurane concentration was reduced to 1.5% at a flow rate of 1 L/min. Measurements were taken before LAD occlusion and after 8 weeks of reperfusion. Ejection fraction (EF, cardiac output/LV diastolic volume) was determined using VINNO LV function software, version VINNO6. Fractional shortening (FS), which takes into consideration one 2D cross-section of the heart, and the LVID at systole and diastole (LVIDs and LVIDd, respectively), were measured.

### 4.7. Assessment of Cardiac Function via Pressure-Volume Loop

Pressure-volume (PV) loop measurements were made using a PV-loop system (SPR-838, Millar Instruments, Houston, TX, USA). An animal under inhalation anesthesia had its end-diastolic pressure (EDP) measured by inserting a PV catheter into the LV through the right carotid artery. Once a steady state was reached, vena cava occlusion was performed to assess LV stiffness and diastolic dysfunction.

### 4.8. Histological Analysis

The slides were stained with Masson’s trichrome stain Kit (Masson Trichrome stain, BBC Biochemical, Mount Vernon, NY, USA) and Picrosirius Red Stain Kit (Picrosirius Red stain, Polysciences, Warrington, PA, USA) for histopathological comparisons. The stained slides were examined under light microscopy (EVOSTM M5000, Thermo Fisher Scientific, Bothell, WA, USA).

### 4.9. Western Blot Evaluation

Heart tissue was homogenized with a lysis buffer containing a protease inhibitor cocktail (K272; Biovision, Inc., Milpitas, CA, USA) and a phosphatase inhibitor cocktail (P5726, Sigma-Aldrich, St. Louis, MO, USA). The homogenates were separated, and the supernatant proteins (30–40 μg/lane) were separated by 10% SDS-PAGE and transferred onto a nitrocellulose membrane (AmershamTM ProtranTM, Buckinghamshire, UK) using a Mini-Protean II instrument (Bio-Rad, Hercules, CA, USA). The membranes were washed with TBS-T [10 mM Tris-HCl (pH 7.6), 150 mM NaCl, 0.05% Tween-20] and blocked with 5% BSA or nonfat milk powder in 1 × TBS-T for 2 h. Then, the membranes were incubated overnight at 4 °C with the appropriate primary antibodies against JAK, p-JAK, STAT3, p-STAT3, caspase-3, Bcl-2, Bax, or β-actin (1:1000 in TBS-T). Protein expression levels were analyzed using a Chemidoc image analyzer (iBright FL 100, Thermo Fisher Scientific, Waltham, MA, USA).

### 4.10. Statistical Analysis

All the experiments were repeated at least three times. Statistical analysis was performed using a *t*-test for multiple comparisons, and *p* < 0.05 was considered statistically significant. Data from echocardiography and PV-loop measurements were analyzed using repeated measures. Student’s *t*-test with the Student’s post-test.

## 5. Conclusions

In conclusion, the findings of this study demonstrate that DHT treatment significantly improves cardiac dysfunction in rats with MI-induced heart failure. The cardioprotective effects of DHT are evident through its ability to enhance cardiac function, with improvements in EF by up to 13% and FS by up to 22%. Notably, the lower dose of DHT (100 mg/kg) showed superior efficacy in some measures, such as EF improvement. Additionally, DHT treatment led to a substantial 63.3% reduction in infarct size and a 16.7% decrease in heart weight. Heart failure biomarkers were also significantly reduced, with LDH levels decreasing by 37.6% and CK-MB levels by 47.6%. Histological analysis further revealed a 50% reduction in left ventricular fibrosis. These effects are mediated, at least in part, by the inhibition of the JAK/STAT3 signaling pathway, with significant reductions in phospho-JAK, phospho-STAT3, Bax/Bcl-2, and caspase-3 expression. These results highlight that DHT has therapeutic potential, particularly at lower doses, for managing MI and heart failure, warranting further investigation into its molecular mechanisms and clinical applicability.

## Figures and Tables

**Figure 1 pharmaceuticals-17-01132-f001:**
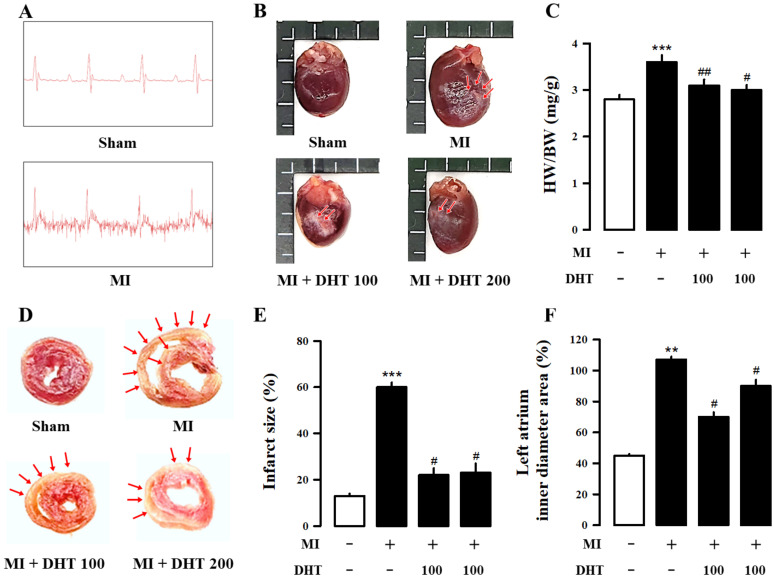
Evaluation of the impact of DHT on cardiac morphology in rats with MI-induced heart failure. (**A**) Representative ECG images confirming the results of myocardial infarction surgery in rats with heart failure. (**B**) Representative images of cardiac tissue TTC staining showing nonischemic areas (red-colored regions) and infarct portions (pale-colored regions indicated by red arrows) in shortened sections from the cardiac apex. (**C**) Quantification of the infarct area in the cardiac tissue. (**D**) Representative images of cardiac chambers for each group. (**E**) Bar graph illustrating the ratios of HW/BW. (**F**) Bar graph displaying the left ventricle inner diameter area percentage. ECG, electrocardiogram; TTC, triphenyltetrazoliumchloride; MI, myocardial infarction; DHT, dohongsamul-tang; HW, heart weight; BW, body weight. Data values are presented as means ± SE. ** *p* < 0.01, *** *p* < 0.01 vs. Sham; # *p* < 0.05, ## *p* < 0.01 vs. MI group.

**Figure 2 pharmaceuticals-17-01132-f002:**
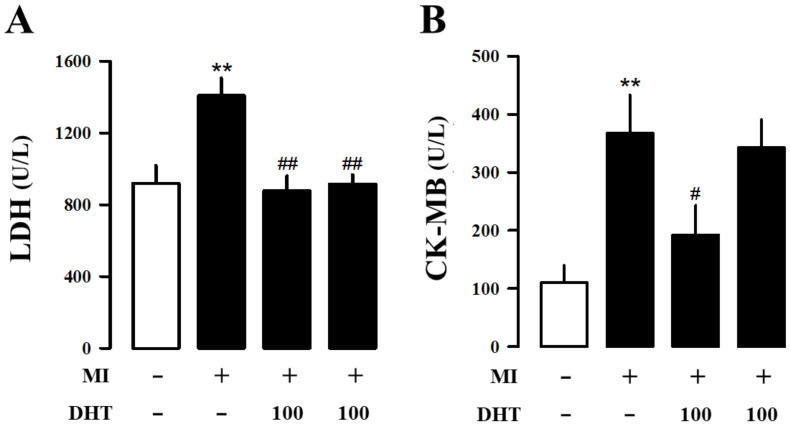
Effects of DHT on hematology analysis in rats with MI-induced heart failure. Analysis of (**A**) LDH and (**B**) CK-MB, which are biomarkers used for assessing heart failure, in plasma. MI, myocardial infarction; DHT, dohongsamul-tang; LDH, lactate dehydrogenase; CK-MB, creatine kinase myocardial band. Data values are presented as means ± SE; n = 8~10. ** *p* < 0.01 vs. Sham; # *p* < 0.05; ## *p* < 0.01 vs. MI group.

**Figure 3 pharmaceuticals-17-01132-f003:**
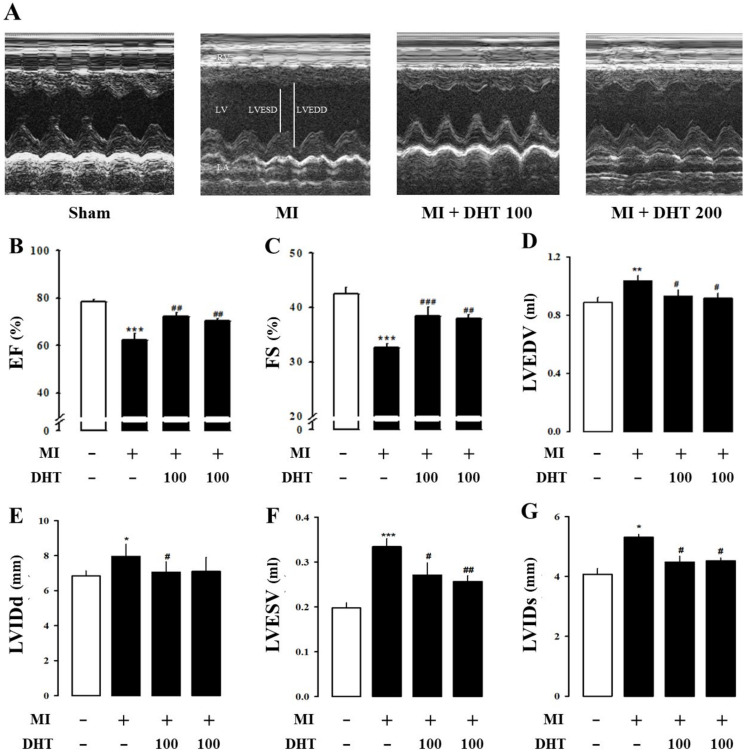
Evaluation of the impact of DHT on echocardiographic parameters in rats with MI-induced heart failure. (**A**) Representative images of M-mode echocardiography obtained from each experimental group. Assessment of DHT has effects on (**B**) EF, (**C**) FS, (**D**) LVEDV, (**E**) LVIDd, (**F**) LVESV, and (**G**) LVIDs. MI, myocardial infarction; DHT, dohongsamul-tang; EF, ejection fraction; FS, fractional shortening; LVEDV, left ventricular end diastolic volume; LVIDd, left ventricular internal diameter at end diastole; LVESV, left ventricular end systolic volume; LVIDs, left ventricular internal diameter at end systole. Data values are presented as means ± SE. * *p* < 0.05, ** *p* < 0.01, *** *p* < 0.01 vs. Sham; # *p* < 0.05, ## *p* < 0.01, ### *p* < 0.001 vs. MI group.

**Figure 4 pharmaceuticals-17-01132-f004:**
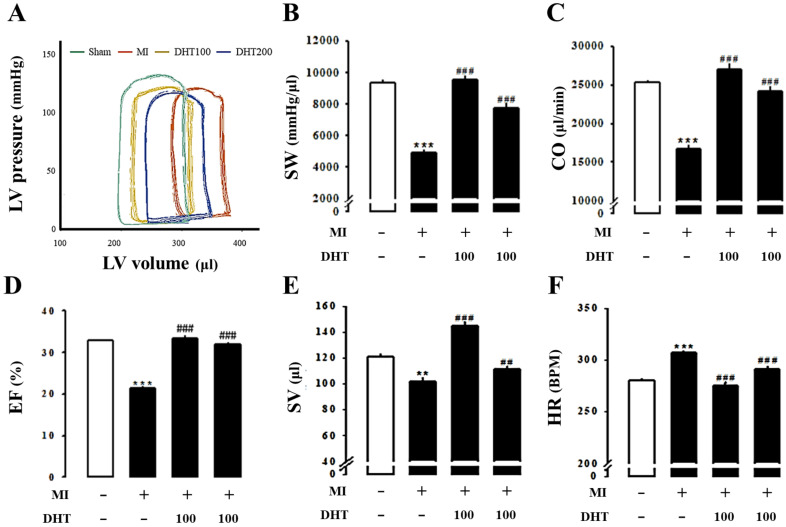
Evaluation of the impact of DHT on pressure-volume loops-derived left ventricle function indices in rats with MI-induced heart failure. (**A**) Representative pressure-volume loops recorded using the Millar pressure-volume catheter system in rats with MI-induced heart failure. Comparison of (**B**) SW, (**C**) CO, (**D**) EF, (**E**) SV, and (**F**) HR among the experimental groups. MI, myocardial infarction; DHT, dohongsamul-tang; SW, stroke work; CO, cardiac output; EF, ejection fraction; SV, stroke volume; HR, heart rate. Data values are presented as means ± SE. ** *p* < 0.01, *** *p* < 0.001 vs. Sham; ## *p* < 0.01, ### *p* < 0.001 vs. MI group.

**Figure 5 pharmaceuticals-17-01132-f005:**
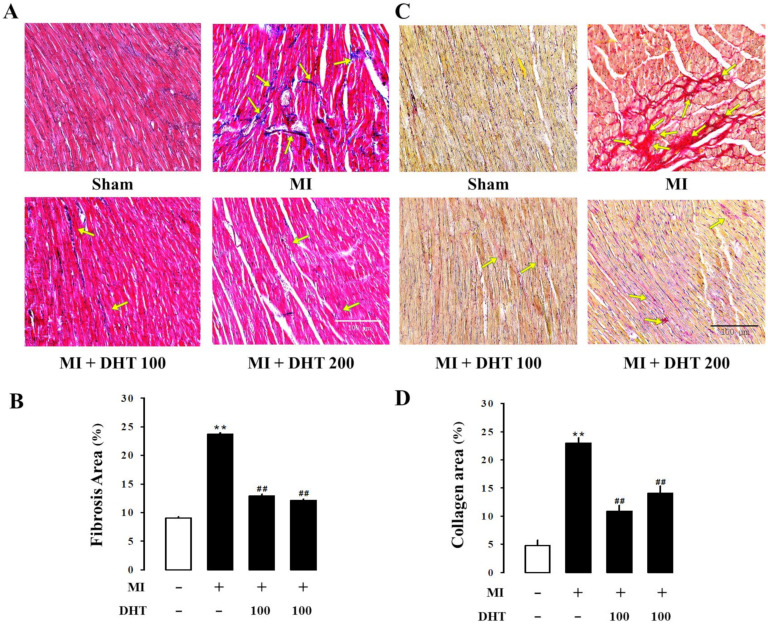
Evaluation of the effect of DHT on cardiac morphology and fibrosis in rats with MI-induced heart failure. Representative images of cardiac fibrosis analysis using (**A**) Masson’s trichrome staining (collagen fibers stained blue) and (**C**) picrosirius red staining (collagen fibers stained red) of the myocardium in rats with MI-induced heart failure (magnification 400×, scale bar: 100 µm). (**B**) The bar graph illustrates the extent of fibrosis in each experimental group, corresponding to (**A**) Masson’s trichrome staining. (**D**) The bar graph illustrates the extent of fibrosis in each experimental group, corresponding to (**C**) picrosirius red staining. Yellow arrows indicate the areas of fibrosis severity. MI, myocardial infarction; DHT, Dohongsamul-tang. Data values are presented as means ± SE. ** *p* < 0.01 vs. Sham; ## *p* < 0.01 vs. MI group.

**Figure 6 pharmaceuticals-17-01132-f006:**
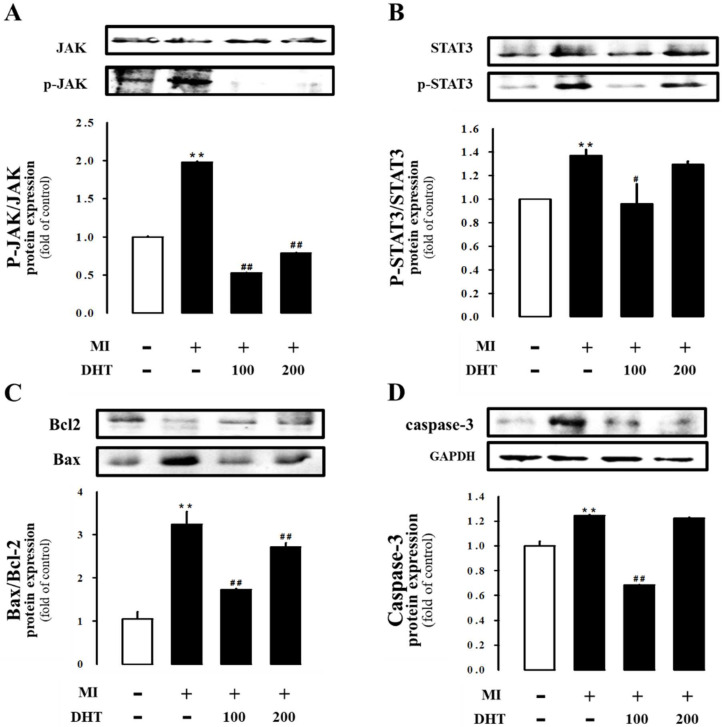
Evaluation of the effect of DHT on the expression of the JAK/STAT3 pathway in rats with MI-induced heart failure. DHT attenuates apoptosis in MI-induced heart failure through the JAK/STAT3 pathway. Representative Western blot bands depicting the expression of (**A**) JAK and p-JAK, (**B**) STAT3 and p-STAT3, (**C**) Bax and Bcl-2, or (**D**) caspase-3, normalized to GAPDH. The bar graph illustrates the protein levels normalized to the sham group in each experimental group. MI, myocardial infarction; DHT, dohongsamul-tang; JAK, Janus kinase; p-JAK, phosphorylated JAK; STAT, signal transducer and activator of transcription; p-STAT, phosphorylated STAT; Bcl-2, B-cell lymphoma protein 2; Bax, Bcl-2-associated X; GAPDH, glyceraldehyde 3-phosphate dehydrogenase. Data values are presented as means ± SE. ** *p* < 0.01 vs. Sham; # *p* < 0.05; ## *p* < 0.01 vs. MI group.

**Table 1 pharmaceuticals-17-01132-t001:** Prescription of dohongsamul-tang (DHT).

Scientific Name	Quantity (g)
*Angelica gigas* Nakai	60
*Prunus persica* Batsch	60
*Rehmannia glutinosa* (Gaertn.) DC.	30
*Paeonia lactiflora* Pallas	30
*Cnidii officinale* Makino	30
*Carthamus tinctorius* Linné	30
Totalily	240

**Table 2 pharmaceuticals-17-01132-t002:** Chemical composition of DHT.

Compounds	Adduct	Rt (min)	Observed *m*/*z*	Fragment Ion
Decursin	[M + H]+	9.72	329.1393	229.0863
Nodakenin	[M + H]+	2.44	409.1504	247.0969
Phenylalanine	[M + H]+	0.78	166.0869	120.0814
Tryptophan	[M + H]+	0.85	205.0979	188.0711
Valine	[M + H]+	13.94	118.0869	72.0808
Amygdalin	[M − H]−	1.14	456.1533	89.0232
Catalpol	[M − H]−	0.76	361.1158	97.0282
Ferulic acid	[M − H]−	2.76	193.0509	134.0365
Glucose	[M − H]−	0.73	179.0559	59.0125
Oxypaeoniflorin	[M − H]−	0.94	495.1530	137.0237
Paeoniflorin	[M + FA − H]−	1.64	525.1639	121.0286
	[M − H]−	1.64	479.1579	
Safflomin A	[M − H]−	1.77	611.1647	325.0737
Stachyose	[M − H]−	0.70	665.2178	383.1152
Sucrose	[M − H]−	0.71	341.1101	179.0553

**Table 3 pharmaceuticals-17-01132-t003:** Effect of DHT on cardiac function in MI-induced heart failure rats.

Group.	LVIDd (mm)	LVIDs (mm)	LVPWd (mm)	LVPWs (mm)
Sham	6.84 ± 0.3	4.07 ± 0.2	2.29 ± 0.2	3.59 ± 0.1
MI	7.96 ± 0.9 *	5.31 ± 0.1 *	1.82 ± 0.1 *	3.02 ± 0.2 *
MI + DHT 100	7.06 ± 0.6 ^#^	4.48 ± 0.2 ^#^	2.19 ± 0.1 ^#^	3.46 ± 0.1 ^#^
MI + DHT 200	7.34 ± 0.8	4.52 ± 0.1 ^#^	2.12 ± 0.1 ^#^	3.36 ± 0.1 ^#^

Cardiac function changes. Transthoracic echocardiography evaluation of Representative tracings of echocardiography in each group. DHT, dohongsamul-tang; MI, myocardial infarction; LVIDd, left ventricular internal dimension at end-diastole; LVIDs, left ventricular internal dimension at end-systole; LVPWd, left ventricular posterior wall thickness at end-diastole; LVIDs, left ventricular internal dimension at end-systole. * *p* < 0.05 vs. Sham; # *p* < 0.05 vs. MI group.

## Data Availability

The datasets used and/or analyzed during the current study are available from the corresponding author on reasonable request.

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
