# Peer review of "The Cardioprotective Potential of Herbal Formulas in Myocardial Infarction-Induced Heart Failure through Inhibition of JAK/STAT3 Signaling and Improvement of Cardiac Function"

_pharmaceuticals, 2024, doi:10.3390/ph17091132_

Round 1

Reviewer 1 Report

Comments and Suggestions for Authors

The article submitted for review concerns the cardioprotective potential of herbal formulas (Dohongsamul-tang), in myocardial infarction-induced heart failure through inhibition of JAK/STAT3 signaling and improvement of cardiac function. The manuscript is a full article examining whether DHT exhibits a protective effect against cardiac dysfunction and fibrosis in rats with heart failure resulting from myocardial infarction. The research also assessed the potential relationship between the cardioprotective actions of DHT and JAK/STAT3 signaling in myocardial infarction (MI) animal model. The article is quite well written, but there are a few points that could make it better. The proposals and main comments to the Authors are listed below.

1.       The “Abstract” section is definitely too long. It should contain the most important achievements of this research and its key (quantitative) results. Meanwhile, the information contained in Lines 17 to 25 would fit more into an “Introduction” than a “Abstract” section, the description of the purpose and course of the research (Lines from 25 to 30) should rather be included in section 2. "Materials and methods", and finally the information included in Lines from 31 to 38 is a rather general discussion of the results. "Abstract" section should include key results and the applicability of the research conducted should be emphasized. Please provide the key (quantitative) results of the research carried out.

2.       More specific terms should be provided in the "keywords" section

3.       In the "Introduction" section, the information regarding Dohongsamul-tang should be expanded, taking into account literature (quantitative) data on bioactive ingredients of functional importance contained in this herbal mixture.

4.       Table 1 contains information in Chinese, it should be replaced with terms in English that are understandable to everyone.

5.       Literature reference (item 11, Line 77) provided incorrectly. It should be given in square brackets, not in superscript.

6.       In the “Materials and methods” section, section 2.2. Myocardial infarction-induced heart failure animal model” the number of animals used in the research and the size of individual experimental groups should be provided. Additionally, it is necessary to provide the consent number of the bioethics committee to conduct research involving animals.

7.       In section 3. "Results", subsection 3.1. “Chemical Characterization and Composition of DHT”: Lines from 195 to 197 - the term "prominent bioactive constituents" in relation to glucose or sucrose is probably an exaggeration. These are not bioactive ingredients...

8.       The readability of Figure 1 should be improved, in particular Figures 1 B (images of cardiac tissue) and 1 D (cardiac chambers).

9.       The readability of Figure 3 should be improved, in particular Figure 1 A (M-mode echocardiography).

10.   The readability of Figure 5 should be improved, in particular Figure 1 A (M-mode echocardiography).

11.   The readability, resolution and size of Figure 5 should be improved, in particular histological images. Additionally, the caption for this Figure 5 is incorrect, because the histological images are shown in figures Aa and Ba, not Bb (Bb is a bar graph).

12.   The weakness of the manuscript is the “Discussion” section. Considering that the results obtained are very interesting, this part of the manuscript should be slightly expanded. The references used are appropriate, but given the existing data in the literature, they are very sparse. Increasing the number of references would perhaps help better substantiate the relevance of the presented results in the context of the existing body of research and provide a more comprehensive review of the literature. It would also be beneficial to include more recent research and a wider range of sources to compare and contrast with current findings.

13.   In the "Conclusions" section, please add quantitative results. The novelty and applicability of the obtained research results should be emphasized.

Author Response

[Reviewer 1]

The article submitted for review concerns the cardioprotective potential of herbal formulas (Dohongsamul-tang), in myocardial infarction-induced heart failure through inhibition of JAK/STAT3 signaling and improvement of cardiac function. The manuscript is a full article examining whether DHT exhibits a protective effect against cardiac dysfunction and fibrosis in rats with heart failure resulting from myocardial infarction. The research also assessed the potential relationship between the cardioprotective actions of DHT and JAK/STAT3 signaling in myocardial infarction (MI) animal model. The article is quite well written, but there are a few points that could make it better. The proposals and main comments to the Authors are listed below.

  1. The “Abstract” section is definitely too long. It should contain the most important achievements of this research and its key (quantitative) results. Meanwhile, the information contained in Lines 17 to 25 would fit more into an “Introduction” than a “Abstract” section, the description of the purpose and course of the research (Lines from 25 to 30) should rather be included in section 2. "Materials and methods", and finally the information included in Lines from 31 to 38 is a rather general discussion of the results. "Abstract" section should include key results and the applicability of the research conducted should be emphasized. Please provide the key (quantitative) results of the research carried out.

→ Thank you for your feedback. We have shortened the Abstract to focus on the most important achievements and key quantitative results of our research. The information that was better suited for the Introduction and Methods sections has been relocated accordingly. Additionally, we have included the key quantitative data to emphasize the specific outcomes of our study.

  1. More specific terms should be provided in the "keywords" section

→ Thank you for your suggestion. We have revised the "Keywords" section to include more specific terms that better reflect the focus of our research. The updated keywords are: Korean Herbal Formulas; Dohongsamul-tang; Myocardial infarction; Heart failure; Cardiac Func-tion; Cardioprotection

  1. In the "Introduction" section, the information regarding Dohongsamul-tang should be expanded, taking into account literature (quantitative) data on bioactive ingredients of functional importance contained in this herbal mixture.

→ In the "Introduction" section, we have expanded the information on Dohongsamul-tang by incorporating details about the bioactive compounds contained in this herbal mixture, supported by quantitative data from recent literature.

  1. Table 1 contains information in Chinese, it should be replaced with terms in English that are understandable to everyone.

→ We apologize for the oversight. The terms in Table 1 have been revised to include only the scientific names of the herbal ingredients, removing any Chinese terms to ensure clarity and accessibility for all readers.

  1. Literature reference (item 11, Line 77) provided incorrectly. It should be given in square brackets, not in superscript.

→ We have revised the reference format to use square brackets as requested. Additionally, the reference numbers have been updated accordingly during the manuscript revision process.

  1. In the “Materials and methods” section, section 2.2. Myocardial infarction-induced heart failure animal model” the number of animals used in the research and the size of individual experimental groups should be provided. Additionally, it is necessary to provide the consent number of the bioethics committee to conduct research involving animals.

→ We have added the number of animals used, the size of the experimental groups, and the bioethics committee approval number to the “Materials and Methods” section. This study was approved by the Institutional Ethics Committee for Animal Experimentation of Wonkwang University (WKU20-23).

  1. In section 3. "Results", subsection 3.1. “Chemical Characterization and Composition of DHT”: Lines from 195 to 197 - the term "prominent bioactive constituents" in relation to glucose or sucrose is probably an exaggeration. These are not bioactive ingredients.

→ Thank you for your valuable feedback. We appreciate your observation regarding the classification of glucose and sucrose as bioactive constituents. In response, we have revised subsection 3.1 to more accurately reflect the distinction between bioactive compounds and other identified substances in DHT. Specifically, glucose and sucrose are present in the formulation but are not considered bioactive ingredients in the context of therapeutic efficacy. We have clarified that the prominent bioactive components of DHT include nodakenin, decursin, amygdalin, catapol, oxypaeoniflorin, paeoniflorin, ferulic acid, and safflomin A, which are derived from the individual herbs in the formula. Thank you for helping us improve the accuracy of our manuscript.

  1. The readability of Figure 1 should be improved, in particular Figures 1 B (images of cardiac tissue) and 1 D (cardiac chambers).

→ We have enhanced the readability of Figure 1, particularly Figures 1B and 1D, by adjusting the image quality and labels for better clarity.

  1. The readability of Figure 3 should be improved, in particular Figure 1 A (M-mode echocardiography).

→ The readability of Figure 3 has been improved by increasing the size of the images and reorganizing the graphs. Additionally, the clarity of Figure 1A has been enhanced.

  1. The readability of Figure 5 should be improved, in particular Figure 1 A (M-mode echocardiography).

→ It seems you were referring to Figure 4's 1A image. We have made the necessary revisions, improving the overall quality of the graphs and enhancing the clarity of the image.

  1. The readability, resolution and size of Figure 5 should be improved, in particular histological images. Additionally, the caption for this Figure 5 is incorrect, because the histological images are shown in figures Aa and Ba, not Bb (Bb is a bar graph).

→ The readability, resolution, and size of Figure 5 have been improved. The caption numbering has been corrected, and we have also revised the figure legends accordingly.

  1. The weakness of the manuscript is the “Discussion” section. Considering that the results obtained are very interesting, this part of the manuscript should be slightly expanded. The references used are appropriate, but given the existing data in the literature, they are very sparse. Increasing the number of references would perhaps help better substantiate the relevance of the presented results in the context of the existing body of research and provide a more comprehensive review of the literature. It would also be beneficial to include more recent research and a wider range of sources to compare and contrast with current findings.

→ Following your suggestions, we have added additional references and made further revisions to the "Discussion" section to provide a more thorough review of the literature. The updated content is highlighted in red for your convenience.

  1. In the "Conclusions" section, please add quantitative results. The novelty and applicability of the obtained research results should be emphasized.

→ Thank you for your valuable feedback. In response to your suggestion, we have revised the "Conclusions" section to include quantitative results that emphasize the novelty and applicability of our research findings. The enhanced conclusion now highlights the significant improvements in cardiac function, reductions in biomarkers, and inhibition of key signaling pathways associated with DHT treatment, thereby underscoring its therapeutic potential for managing MI and heart failure.

# Thank you for Reviewer 1's valuable feedback and for taking the time to review our manuscript.

Reviewer 2 Report

Comments and Suggestions for Authors

The manuscript entitled: "The Cardioprotective Potential of Herbal Formulas in Myocardial Infarction-Induced Heart Failure through Inhibition of JAK/STAT3 Signaling and Improvement of Cardiac Function" was revised and in general view. It was very well presented and followed, and it provided evidence of the protective propriety of the DHT treatment in a model of MI. A "clean" model of the disease. However, that manuscript has some pitfalls that authors have to discuss in order to represent this model, which happens in MI in humans, which is added metabolic disorders such as high cholesterol and the presence of atherosclerosis. For these reasons, the model used is a "clean" one that could be distant from what happens in human patients.

Authors must attend observations:

1. The abstract is so long. Revise

2. Could the authors describe in more detail how much the MI induced by the ligation is similar to the block of blood flow due to obstruction by atherosclerosis or similar since this type of MI is more local than the one induced in the model? REvise.

3. Reference 20. This reference refers to Latent-Transforming Growth Factor ß-binding Protein 2, which Accelerates Cardiac Fibroblast Apoptosis by Regulating the Expression and Activity of Caspase-3. The authors say, "under stressful conditions such as MI," and I think it is not a reference to support it. Revise.

4. How do authors explain the treatment with 200 of DHT, can not low expression of Caspase-3, and are high the level of Bax/Bcl-2 expression? Also, p-STAT/STAT? The high levels of expression of these genes are also correlated with high CK-MIB levels.  How could authors explain why the nonlineal response to the hearts with MI is not much better recovered with 200 of DHT? The authors have to discuss these model responses very well. Revise.

5. In Figure 7, the authors must include the increase of collagen fibers in MI.

6. "The cardioprotective effects of DHT" in conclusion is at one DHT concentration

7. "Further research is warranted to fully understand the molecular mechanisms involved and to explore the clinical potential of DHT and its active components in the management of MI and heart failure"; with this in consideration, authors must discuss there is a maximum of DHT concentration for promissory use, derived from this study in the animal model. 

Author Response

[Reviewer 2]

Comments and Suggestions for Authors

The manuscript entitled: "The Cardioprotective Potential of Herbal Formulas in Myocardial Infarction-Induced Heart Failure through Inhibition of JAK/STAT3 Signaling and Improvement of Cardiac Function" was revised and in general view. It was very well presented and followed, and it provided evidence of the protective propriety of the DHT treatment in a model of MI. A "clean" model of the disease. However, that manuscript has some pitfalls that authors have to discuss in order to represent this model, which happens in MI in humans, which is added metabolic disorders such as high cholesterol and the presence of atherosclerosis. For these reasons, the model used is a "clean" one that could be distant from what happens in human patients.

Authors must attend observations:

  1. The abstract is so long.

→ Thank you for your feedback and for helping us improve the clarity of our manuscript. We have revised the abstract to make it more concise, focusing on the key findings and their implications.

  1. Could the authors describe in more detail how much the MI induced by the ligation is similar to the block of blood flow due to obstruction by atherosclerosis or similar since this type of MI is more local than the one induced in the model?

→ Thank you for your question. The myocardial infarction (MI) model used in our study involves left anterior descending (LAD) artery ligation to induce a localized and acute blockage of blood flow. This approach is designed to replicate the effects of sudden and severe coronary occlusion, providing insights into acute myocardial damage and subsequent heart failure. In contrast, myocardial infarction due to atherosclerosis typically results from a gradual and progressive narrowing of the coronary arteries due to plaque buildup. This type of obstruction is more chronic and diffuse, leading to partial or intermittent blood flow reduction before complete blockage occurs. Atherosclerosis-induced MI often affects multiple regions of the heart over time and may not present as a single, acute infarct. While the LAD ligation model effectively simulates the immediate consequences of severe coronary obstruction, it does not fully replicate the chronic, progressive nature of atherosclerosis. To model atherosclerosis more closely, additional approaches that simulate gradual arterial stenosis would be required. We have included this explanation in the [discussion section] to provide a clearer context for the limitations and relevance of the LAD ligation model in the study of myocardial infarction.

  1. Reference 20. This reference refers to Latent-Transforming Growth Factor ß-binding Protein 2, which Accelerates Cardiac Fibroblast Apoptosis by Regulating the Expression and Activity of Caspase-3. The authors say, "under stressful conditions such as MI," and I think it is not a reference to support it.

→ Thank you for your feedback. Considering that the cited reference did not directly support the statement in question, we have replaced it with a more appropriate source (Reference 30).

  1. How do authors explain the treatment with 200 of DHT, can not low expression of Caspase-3, and are high the level of Bax/Bcl-2 expression? Also, p-STAT/STAT? The high levels of expression of these genes are also correlated with high CK-MIB levels.  How could authors explain why the nonlineal response to the hearts with MI is not much better recovered with 200 of DHT? The authors have to discuss these model responses very well.

→ Thank you for your thoughtful question. We appreciate the opportunity to clarify our findings regarding the treatment with 200 mg/kg of DHT. While we observed some promising effects of DHT on cardiac function, the higher dose of 200 mg/kg did not achieve the expected reduction in caspase-3 expression. Instead, we found increased levels of Bax/Bcl-2 ratios and p-STAT/STAT, which were also associated with higher CK-MB levels. This suggests that the higher dose may lead to unintended effects or additional stress on the heart. In our experience with various studies, we have often observed that high doses of certain herbal formulations, including DHT, do not always produce dose-dependent effects. This phenomenon might be due to complex interactions at higher concentrations, which requires further investigation. In the [discussion section], we acknowledge these limitations and emphasize the need for additional research to better understand why high doses might not be as effective and to optimize treatment protocols. Despite these challenges, our study highlights the potential therapeutic benefits of DHT for cardiac injury and heart failure.

  1. In Figure 7, the authors must include the increase of collagen fibers in MI.

→ We have updated Figure 7 to include the increase of collagen fibers in the MI group, as suggested.

  1. "The cardioprotective effects of DHT" in conclusion is at one DHT concentration

→ Thank you for your feedback. We have revised the conclusion to reflect that the cardioprotective effects of DHT were more pronounced at lower concentrations.

  1. "Further research is warranted to fully understand the molecular mechanisms involved and to explore the clinical potential of DHT and its active components in the management of MI and heart failure"; with this in consideration, authors must discuss there is a maximum of DHT concentration for promissory use, derived from this study in the animal model. 

→ Thank you for your valuable feedback. We agree that further research is necessary to explore the optimal concentration of DHT for potential clinical use. As noted, our study suggests that while lower doses of DHT are beneficial, higher doses may not provide additional benefits. In the [discussion section], we have addressed the need for future studies to determine the maximum effective concentration of DHT for clinical applications. This will help optimize dosing strategies and fully understand the therapeutic potential of DHT in managing MI and heart failure.

# Thank you for Reviewer 2's valuable feedback and for taking the time to review our manuscript.
